# TGFβ/BMP Signaling Pathway in Cartilage Homeostasis

**DOI:** 10.3390/cells8090969

**Published:** 2019-08-24

**Authors:** Nathalie G.M. Thielen, Peter M. van der Kraan, Arjan P.M. van Caam

**Affiliations:** Experimental Rheumatology, Radboud University Medical Center, Geert Grooteplein 28, 6525 GA Nijmegen, The Netherlands

**Keywords:** transforming growth factor β, bone morphogenetic proteins, osteoarthritis, cartilage, SMADs, aging, joint loading, inflammation, linker modifications

## Abstract

Cartilage homeostasis is governed by articular chondrocytes via their ability to modulate extracellular matrix production and degradation. In turn, chondrocyte activity is regulated by growth factors such as those of the transforming growth factor β (TGFβ) family. Members of this family include the TGFβs, bone morphogenetic proteins (BMPs), and growth and differentiation factors (GDFs). Signaling by this protein family uniquely activates SMAD-dependent signaling and transcription but also activates SMAD-independent signaling via MAPKs such as ERK and TAK1. This review will address the pivotal role of the TGFβ family in cartilage biology by listing several TGFβ family members and describing their signaling and importance for cartilage maintenance. In addition, it is discussed how (pathological) processes such as aging, mechanical stress, and inflammation contribute to altered TGFβ family signaling, leading to disturbed cartilage metabolism and disease.

## 1. Introduction

The transforming growth factor β (TGFβ) family of polypeptide growth factors controls development and homeostasis of many tissues, including articular cartilage. Articular cartilage is the connective tissue covering joint surfaces and is a type of hyaline cartilage. This tissue is key in facilitating movement with its smooth lubricated surface, and it functions as a shock absorber to disperse forces acting upon movement with its physical properties. Articular cartilage is of mesodermal origin and, remarkably for a tissue, is neither innervated nor does it contain blood vessels to supply oxygen or nutrients [1,2]. The only cell type present in this tissue is the chondrocyte, which is surrounded by a large amount (up to 98% of cartilage volume) of extracellular matrix (ECM) [1,3]. The ECM is a highly organized network of hyaluronan, proteoglycans, and collagens. These collagens provide (tensile) strength and shape to the tissue [4], and they mainly are (up to 90%) collagen type II (COL2A1) fibrils [5]. Hyaluronan is a glycosaminoglycan (GAG) (i.e., a long, unbranched polysaccharide), which forms spontaneously aggregating networks to which other biomolecules such as proteoglycans can bind [6]. Proteoglycans are proteins glycosylated with sulfated GAGs. The most abundant proteoglycan (in mass) in articular cartilage is aggrecan (ACAN), and this proteoglycan forms large aggregates with hyaluronan. Because ACAN is heavily glycosylated with negatively charged sulfated GAGs, these aggregates generate a (static) charge density. This charge density generates the force required to counteract compressive forces during movement via attraction of solutes, which generate osmotic swelling pressure, and via electrostatic repulsion if aggrecan molecules come in close proximity [7].

Articular cartilage is a highly structured tissue in which form fits function. Five distinct zones can be observed: the lamina splendens, the superficial zone, the middle zone, the deep zone, and the calcified zone (Figure 1). The lamina splendens consists of collagen fibrils oriented in parallel to the surface and high levels of hyaluronan and the lubricating proteoglycan lubricin (PRG4), but it contains no cells [8,9,10]. These characteristics provide a smooth surface with a low friction coefficient. Below the lamina splendens lies the superficial zone, which contains a relatively high number of cells with flattened morphology oriented in parallel to the surface, which produce high amounts of hyaluronan and PRG4 [8,10]. The collagen fibrils here are tightly packed and also oriented in parallel to the surface. This orientation counteracts osmotic swelling pressure of the tissue below [9]. Beneath the superficial zone lies the middle zone, which represents 40%–60% of articular cartilage [2]. Chondrocytes in this zone are rounded but sparse. The collagen fibers are oriented isotropically to the cartilage surface [9], which allows for efficient collapse of the tissue upon compression to dissipate the energy of the impact. The collapse of this layer is accompanied by displacement and loss of water. The more this layer is compacted, the more resistance to further collapse is increased [1]. This resistance to further compression is due to an increase in electrostatic repulsion between proteoglycans and also to an increase in hyaluronan viscosity because of an increase in concentration of this molecule [11]. Underneath the middle zone lies the deep zone (±30%). In the deep zone, chondrocytes are organized in columns placed between collagen fibrils oriented perpendicular to the cartilage surface [6,9]. Furthermore, this zone contains relatively the highest levels of aggrecan of all zones [1]. The high levels of aggrecan restrict water flow and thereby increase the compressive stiffness of cartilage. The collagen fibers are oriented to withstand the swelling pressure generated by these proteoglycans and to withstand sheer stresses generated by compression of the layers above. Part of the deep zone, below the tidemark, contains calcium salts in the ECM and is, thus, called the calcified zone [2]. The calcified zone forms the interface between bone and cartilage and anchors cartilage to the bone. This zone contains hypertrophic-like chondrocytes which resemble chondrocytes undergoing endochondral ossification [12].

The importance of articular cartilage for joint function is illustrated by the effects of its loss, which leads to disability and pain, for example, in the word’s most common joint disease osteoarthritis (OA). Degeneration and loss of articular cartilage are the consequence of an imbalance in anabolic (e.g., ECM production) and catabolic (e.g., ECM degradation) processes. Chondrocytes, as the only cell type present, are essential for balancing these processes. Therefore, chondrocyte metabolism is a key regulator of cartilage homeostasis and joint health. Many factors can regulate chondrocyte homeostasis, like hydrostatic pressure, tensile strain, and proinflammatory cytokines, but various growth factors, including those of the TGFβ family, play a key role in this. Growth factors modulate chondrocyte metabolism, differentiation, proliferation, and survival, and they regulate ECM production and turnover. This review will address the pivotal role of the TGFβ family in chondrocyte (and thus cartilage) biology by listing several TGFβ family members and describing their importance. We will discuss how these members signal and how this signaling impacts cartilage. Furthermore, we will address how (pathological) processes such as aging, mechanical stress, and inflammation influence TGFβ signaling in chondrocytes, leading to an altered metabolism and disease.

## 2. The Transforming Growth Factor Β (TGFβ) Family and Its Signaling

The TGFβ family consists of over 30 members, including the TGFβs, activins, bone morphogenetic proteins (BMPs), and growth/differentiation factors (GDFs). These factors share a characteristic quaternary dimer structure [13] and signal via formation of heteromeric complexes between two types (type I and type II) of serine threonine-protein kinase receptors [14].

There are seven type I and five type II receptors (Figure 2). Because every growth factor recruits a specific, but not necessarily unique, receptor complex, this choice in receptors allows for differences in growth factor sensitivity. In addition, multiple type III receptors have been characterized that facilitate the interaction between ligand and receptor and can stabilize receptor complexes. For example, for TGFβ betaglycan [15,16] and endoglin [17], and for BMPs the repulsive guidance molecule (RGM) family members, RGM domain family member A (RGMA), dragon, and hemojuvelin have been identified as co-receptors [18]. Also, inhibitory (pseudo) receptors have been identified, which bind ligands but do not initiate signaling (e.g., CD109 for TGFβ signaling [19] and BMP and activin membrane bound inhibitor (BAMBI) for BMP and activin signaling [20]). After formation of functional receptor complexes, TGFβ family members can activate multiple intracellular signaling pathways, which are categorized as either receptor SMAD (R-SMAD)-dependent or -independent, but R-SMAD-dependent signaling is regarded as the unique and canonical signaling pathway of TGFβ family members.

### 2.1. R-SMAD-Dependent Signaling

R-SMADs are proteins approximately 50 kDa in size that, upon receptor-mediated activation by carboxy terminal phosphorylation, migrate to the nucleus and function as transcription factors. Two groups of R-SMADs can be distinguished based upon structural and functional homology: one group consisting of SMAD1, SMAD5, and SMAD9 (also known as SMAD8) and one group containing SMAD2 and SMAD3 [14]. Upon receptor type I activation, these R-SMADs are directly phosphorylated on two serines at their C-terminal SXS motif. Specific ALKs phosphorylate specific SMADs: ALK1, 2, 3, and 6 phosphorylate SMAD1, 5, and 9, whereas ALK4, 5, and 7 phosphorylate SMAD2 and SMAD3. This phosphorylation facilitates complexation of R-SMADs with the common SMAD (co-SMAD), SMAD4, which promotes nuclear entry and accumulation. Inside the nucleus, this SMAD complex directly binds DNA, although with low affinity, and recruits transcription factors for \high-affinity interactions with DNA. The location where SMAD complexes bind DNA is dependent on characteristic SMAD-binding motifs present in the DNA sequence, which differ for SMAD3/4 and SMAD1/5/9 (e.g., GTCTAGAC for SMAD2/3/4 complexes [21] and GCCGnCGC for SMAD1/5/9/4 complexes [22]). Furthermore, the interaction between SMADs and DNA is regulated by the presence of master regulator transcription factors [23]. Master regulators vary per tissue, and an important master regulator of chondrocyte phenotype is SRY-box9 (SOX9), for example, which closely interacts with SMAD3/4 on the collagen type II promotor [24].

Two important target genes of R-SMAD signaling are the inhibitory-SMADs (I-SMAD): *SMAD6* and *SMAD7* [25,26]. These I-SMADs function as inhibitors of R-SMAD signaling on multiple levels by inhibiting C-terminal R-SMAD phosphorylation by inducing receptor dephosphorylation and degradation, by inhibiting R-SMAD DNA binding, and by inhibiting complex formation between SMAD4 and R-SMADs. In this way, SMAD6 predominantly inhibits SMAD1/5/9, and SMAD7 predominantly inhibits SMAD2/3 signaling, and both I-SMADs provide cells with an important negative feedback mechanism to inactivate R-SMAD signaling.

A (simplified) overview of the SMAD-dependent signaling pathway is depicted in Figure 2.

### 2.2. SMAD-Dependent Signaling in Chondrocytes and Cartilage Biology

SMAD signaling is essential for the formation and maintenance of healthy cartilage. Both in vivo and in vitro experiments point towards essential, but distinct, roles for SMAD2/3 versus SMAD1/5/9 signaling in chondrocyte biology.

In humans, genetic variation in *SMAD3* is associated with hip and knee OA and total burden of the disease [27,28]. Dominant nonsense, missense, or frameshift mutations in *SMAD3* lead to aneurysms-OA syndrome, a severe condition characterized by cardiovascular anomalies and early onset of OA in multiple joints [29,30]. Mice with a *Smad3* null mutation (*Smad3*^Δexon8/^^Δexon8^) develop normal articular cartilage, but this starts to rapidly degrade one month after birth, resulting in severe OA [31]. This rapid degradation is linked to an increased number of hypertrophic chondrocytes present in both articular and epiphyseal cartilage [31]. These hypertrophic chondrocytes are unable to maintain a healthy articular cartilage matrix and produce collagen type 10 (COL10A1) and matrix metalloprotease 13 (MMP13), a potent cartilage-degrading enzyme. Increased chondrocyte hypertrophy and accelerated cartilage degeneration are also observed in chondrocyte-specific *Smad3*-null mice (*Smad3*^Δexon2 + 3/^^Δexon2 + 3^), which have been made using *Col2-Cre* and floxed *Smad3* to circumvent systemic effects of *Smad3* deletion, confirming the direct role of SMAD3 in these processes [32]. In vitro studies using overexpression or knockout of *SMAD3* also support a role for SMAD3 in inhibiting chondrocyte terminal differentiation, by showing an inhibitory or stimulating effect on chondrocyte hypertrophy, respectively [33,34,35,36]. The anti-hypertrophic effect of SMAD3 is attributed to its interaction with core-binding factor subunit α 1 (CBFA1), also known as runt-related transcription factor 2 (RUNX2) [32,37,38]. RUNX2 is a potent regulator of chondrocyte maturation [39]. For example, overexpression of *Runx2* in the murine teratoma-derived ATDC5 chondrocyte-like cell line induces chondrocyte hypertrophy, whereas inhibition of RUNX2 achieves the opposite [32,38,40,41]. SMAD3 counteracts RUNX2 function by directly binding this transcription factor and recruiting silencing histone class II deacetylases to RUNX2 responsive genes [37] such as *MMP13*, *COL10A1,* and alkaline phosphatase (*ALPL*).

The role of SMAD2 in cartilage biology is less clear than that of SMAD3. In humans, there is no known association between genetic variation in *SMAD2* and OA. *Smad2* knockout animals or hypomorphs are not viable and die early during embryogenesis (≤embryonic day 9 [42,43,44,45]), before the formation of articular joints [46], and are thus not usable to study the role of SMAD2 in articular cartilage. Animals with only one genetic copy of *Smad2* (*Smad2*^+/−^) can be viable [42,47], but no cartilage phenotype has been reported. Cartilage-specific *Smad2* knockout mice (*Smad2^fx/fx^;Col2a1Cre)* have recently been studied [48], and such mice have an elongated hypertrophic zone in their growth plate. This last observation indicates a similar anti-hypertrophic effect of SMAD2 as SMAD3. An in vitro study supports a role for SMAD2 similar to that of SMAD3, because overexpression of a dominant negative variant of *Smad2* enhances, whereas overexpression of wildtype *Smad2* protects against chondrocyte hypertrophy, although less potent than SMAD3 [33].

SMAD1 and SMAD5 seem to have an important, but interchangeable, role in cartilage biology. In contrast, SMAD9 seems to be of little importance for cartilage formation or maintenance. In humans, no association has been reported between genetic variations in *SMAD1*, *SMAD5,* or *SMAD9* and OA, possibly due to redundant roles in cartilage. Like *Smad2* knockout mice, *Smad1* [49] or *Smad5* [50] knockout mice die before the formation of articular joints. Knockout of *Smad9* is not embryonically lethal yet does not seem to affect (epiphyseal) cartilage biology [51]. To study SMAD1 and SMAD5 in chondrogenesis, cartilage-specific knockouts have been made using *Col2-cre* and floxed *Smad1* or *Smad5* [51]. Cartilage-specific removal of either *Smad1* or *Smad5* does not result in a phenotype, nor in combination with whole body *Smad9* knockout. Only when both *Smad1* and *Smad5* are ablated, severe chondrodysplasia can be observed [51]. Additional removal of *Smad9* does not add much to this severe phenotype. Together, these observations indicate that SMAD1 and SMAD5 have an important, but redundant, role in cartilage formation, whereas the role of SMAD9 seems limited [51]. The severe chondrodysplasia is, in part, due to a lack of hypertrophic chondrocytes in the growth plates, suggesting that SMAD1/5 signaling regulates chondrocyte maturation. In vitro studies support a stimulatory role of SMAD1/5 in chondrocyte maturation. For example, overexpression of *Smad6*, an inhibitor of SMAD1/5/9, results in inhibited chondrogenesis in the ATDC5 cell line [52] and delayed chondrocyte hypertrophy [53]. The stimulatory effects of SMAD1/5 on chondrocyte maturation most likely are due to a stimulation of RUNX2 function. Such a stimulatory role of SMAD1/5 on RUNX2 function has been described to be essential for BMP2-induced osteoblast differentiation [54,55,56,57]. Furthermore, in pre-hypertrophic chondrocytes, SMAD1/5-RUNX2 interaction induces the activation of a reporter construct based on the *Col10a1* promoter, the expression of which is an important phenotypic marker of hypertrophy [38,58]. How exactly SMAD1 and SMAD5 stimulate RUNX2 function is yet unclear.

### 2.3. SMAD-Independent Signaling

As mentioned, TGFβ family members can also activate R-SMAD-independent pathways. Two examples are signaling via mitogen-activated protein kinase (MAPK) pathways involving TGFβ-activated kinase 1 (TAK1) or extracellular signal-regulated kinase 1 and 2 (ERK1/2, also known as MAPK3/1). The SMAD-independent signaling pathways are depicted in Figure 2. Of note, these pathways are not unique for TGFβ family signaling but are important signaling mediators of many other growth factors and cytokines as well.

Activation of MAPKs by TGFβ family members relies on adaptor proteins to bridge receptor activity to pathway activation—for example, SHC adaptor protein 1 (SHC1) for ERK [59] and tumor necrosis factor (TNF) receptor-associated factor 6 (TRAF6) for TAK1 activation [60,61]. Receptor-mediated phosphorylation of SHC1 triggers a MAPK signaling cascade via GRB2/SOS, RAS, RAF, and MAP2K1, resulting in activation of ERK1/2 [59]. TRAF6 is activated differently: upon receptor type I/type II complex formation, TRAF6 oligomerization and auto-ubiquitination occurs, triggering complex formation between TAK1 and TGFβ-activated kinase 1/MAP3K7 binding protein 1 (TAB1) via poly-ubiquitination, which allows TAK1 auto-phosphorylation and activity [61]. Alternatively, TAK1 can be activated by E3 ubiquitin-protein ligase XIAP (in complex with TAB1), the expression of which is stabilized by interaction with BMP (and TGFβ) receptors [62,63]. Ultimately, activation of MAPK signaling by TGFβ family members results in activation of JNK and p38 kinases, which directly regulate gene expression by modulating transcription factors such as Jun proto-oncogene (c-JUN), MYC proto-oncogene (c-MYC), activating transcription factor 2 (ATF2), and CCAAT/enhancer-binding protein β (C/EBP-β) [64].

Both ERK and TAK1 contribute to TGFβ family signaling in chondrocytes. Functional inhibition of ERK1 in ATDC5 cells via inhibition of its activating kinase MAP2K1 with the inhibitor U0126 reduces TGFβ-induced *Acan* expression [65]. Functional inhibition of ERK1 in rat chondrocytes with the use of the MAP2K1 inhibitor PD98059 reduces TGFβ-induced proliferation and *Acan* and *Col2a1* expression [66,67]. Furthermore, in both human and bovine primary chondrocytes, use of PD98059 lowers TGFβ-induced mRNA and protein expression of the metallopeptidase inhibitor TIMP3 [68], indicating that ERK activity is a cartilage-protective component of TGFβ signaling in mature chondrocytes. In contrast, during TGFβ-induced chondrogenesis in vitro, use of PD98059 increases *Col2a1* and *Sox9* expression and proteoglycan synthesis in rat, chicken, and human mesenchymal stem cells [69,70,71], indicating that ERK1/2 is an inhibitory component of TGFβ signaling during TGFβ-induced chondrogenesis.

Functional inhibition of TAK1 using a dominant negative form of *Tak1* greatly diminishes the ability of TGFβ and BMP2 to induce *Col2a1* expression in rabbit articular chondrocytes [72]. In *Col2-Cre;Tak1^fl/fl^* mice, *Tak1* deletion in *Col2a1*-expressing cells during embryogenesis results in runting and severe chondrodysplasia because of the reduced chondrocyte proliferation and delayed maturation [73], and these mice die shortly before [73] or after birth [74]. This severe phenotype resembles the phenotype of mice lacking BMP receptor/SMAD(1/5) signaling. Indeed, these mice show impaired BMP-induced R-SMAD and p38 MAPK/JNK/ERK signaling [73,74], confirming that TAK1 is an important component of BMP signaling in cartilage. Postnatal deletion of *Tak1* using inducible *Col2a1-CreER^T2^;Tak1^fl/fl^* mice also results in growth retardation, and these mice have profoundly decreased proteoglycan and COL2A1 production in their articular cartilage [75]. This last observation shows that the importance of TAK1 in cartilage biology is not limited to embryogenesis.

A possible role of TAK1 signaling in chondrocytes is the regulation of SOX9 production. In chondrocytes, deletion of *Tak1* inhibits endogenous and BMP2-induced *Sox9* mRNA expression and protein production, whereas overexpression of *Tak1* enhances SOX9 levels [75]. This observation is reflected in vivo: both articular and epiphyseal chondrocytes from *Col2a1-CreER^T2^;Tak1^fl/fl^* mice show lower SOX9 protein production after deletion of *Tak1* compared to control animals [75]. The induction of *Sox9* expression by TAK1 involves binding of activating transcription factor 2 (ATF2), a downstream target of TAK1 via p38 MAPK, to the *Sox9* promoter [75]. Furthermore, p38 MAPK signaling has also been shown to stabilize *SOX9* mRNA in human chondrocytes, further strengthening the importance of this MAPK pathway in SOX9 regulation [76].

TGFβ family members can also signal via phosphoinositide 3-kinase (PI3K) or via the GTPase RhoA [64], but it is unclear how and to what extent this occurs in chondrocytes. Activation of PI3Ks leads to phosphorylation and activation of AKT serine/threonine kinase 1/2/3 (AKT1/2/3) and of mechanistic target of rapamycin (MTOR) [77]. Recently, it has been shown that mechanistic target of rapamycin complex 1 (mTORC1) activity is required for chondroblast growth and proliferation in the epiphyseal plate of mice, but that its inactivation is required for chondrocyte differentiation [78]. In articular cartilage, MTOR is upregulated during OA [79] and is an inhibitor of chondroprotective autophagy [79,80,81]. In vivo, intra-articular or intra-peritoneal injection of rapamycin (i.e., an inhibitor of MTOR) increases autophagy and reduces cartilage damage in the DMM (surgical destabilization of the medial meniscus) model of OA [80,81]. Furthermore, cartilage-specific *Mtor* knockout mice (*Col2-rt-TA-Cre*; *Mtor*^fl/fl^) are protected against DMM-induced OA [79], confirming that MTOR signaling is deleterious in OA conditions.

In chondrocytes, RhoA activates rho-associated coiled-coil containing protein kinase 1 (ROCK1) [82,83]. In primary human chondrocytes, ROCK inhibition induces *SOX9*, *COL2A1,* and *ACAN* mRNA expressions [76], while in primary murine chondrocytes, ROCK inhibition enhances the output of a SOX9-responsive luciferase assay along with *Col2a1* and *Acan* mRNA expression [83]. Therefore, these studies indicate a negative effect of RhoA/ROCK signaling on chondrocyte phenotype. However, in murine mesenchymal cells during chondrogenic differentiation and chondrocytes cultured in 3D, ROCK1 inhibition negatively affects SOX9 function [83]. This effect has been attributed to direct phosphorylation of SOX9 by ROCK1, leading to nuclear accumulation and transcriptional activity of SOX9 [84]; thus, in contrast to the previous studies, this indicates a positive effect of Rhoa/ROCK signaling on chondrocyte phenotype.

A (simplified) overview of the SMAD-independent signaling pathways is depicted in Figure 3.

## 3. TGFβ Family Members in Cartilage

### 3.1. TGFβ1, TGFβ2, and TGFβ3

The importance of TGFβ in cartilage biology was first observed in 1985 when two proteins were extracted from bone that were capable of inducing chondrogenesis in rat embryonic mesenchymal cells, and therefore were called cartilage inducing factor A and B [85]. Later research revealed that these proteins were identical to TGFβ1 and TGFβ2 [86,87].

In mammals, three TGFβs exist—TGFβ1, TGFβ2, and TGFβ3—encoded by three different genes—*TGFB1*, *TGFB2* and *TGFB3*—and these homologs share a high percentage of amino acid identity (71%–79%). All three TGFβs are produced in inactive forms as homodimers (or more rarely as heterodimers [88]) bound by latency-associated peptide (LAP) and latent TGFβ binding protein (LTBP) [89]. This production in the inactive form separates secretion from activity, an important concept in TGFβ biology. All three forms of TGFβ are produced by chondrocytes [90,91,92] and a large amount (60–200 ng/mL) of inactive TGFβ is bound to the ECM of cartilage [93,94]. Because TGFβ is produced in the inactive form, activation is a crucial step in TGFβ signaling. For this, the noncovalent bond between LAP and TGFβ has to be disrupted. This can occur enzymatically by degradation of LAP by, for example, MMP3 [95], or nonenzymatically by a conformational change in the tertiary structure of LAP. Shearing stress [96] or chemical modification of LAP by reactive oxygen species [97] can induce such a conformational change. In addition, mechanical force as produced by compressive loading was also shown to be an important physiological activator of ECM-bound TGFβ in cartilage [98].

In chondrocytes, TGFβ induces both SMAD2/3 and SMAD1/5/9-dependent signaling via ALK5 and ALK1, respectively, in addition to SMAD-independent signaling [99,100]. Which pathway is activated by TGFβ depends on receptor level expression, co-receptor expression (i.e., endoglin enhances signaling via ALK1), and dose of TGFβ. In chondrocytes, a low dose of TGFβ predominantly signals via pSMAD2/3, whereas at high dosages, pSMAD1/5 signaling becomes more pronounced. Importantly, both pathways have been described to antagonize each other in chondrocytes [99,100].

TGFβ signaling is associated with cartilage ECM production and maintenance. In mice, a single intra-articular injection of rhTGFβ1 or rhTGFβ2 increases proteoglycan synthesis twofold after four days, as measured by ^35^S-sulfate incorporation in patellar articular cartilage [101,102]. Also in vitro, TGFβ enhances proteoglycan production in both chondrocytes and cartilage explants on both mRNA and protein level [103,104,105,106], together with production of other ECM components like COL2A1 [106,107], cartilage oligomeric matrix protein (COMP) [108,109], perlecan [110], fibronectin [111], and hyaluronan [112]. Furthermore, TGFβ is a potent inducer of lubricin (PRG4) in chondrocytes [113], a key lubricating component of synovial fluid. However, in contrast to the previously mentioned studies, inhibitory effects of TGFβ on chondrocyte proliferation, proteoglycan synthesis, and COL2A1 production have also been described both in vitro and in vivo [107,114,115,116,117,118,119]. Possibly, different receptor signals play a role in these apparently contradictory observations.

Furthermore, TGFβ signaling has important anti-inflammatory effects in cartilage. TGFβ1 counteracts proinflammatory IL-1 signaling in vivo [120,121,122] and in vitro [123], and it helps cartilage proteoglycan content recover after inflammation induced depletion [90,124]. How TGFβ exerts its anti-inflammatory effects is not fully elucidated yet, but in primary rabbit articular chondrocytes, both TGFβ1 and TGFβ3 downregulate IL-1 receptor (IL1R1) expression approximately 50% on protein and mRNA level [123,125]. Furthermore, proinflammatory IL-6 signaling is also counteracted by TGFβ by downregulation of IL-6 receptor levels in human and bovine chondrocytes [126]. In macrophages, TGFβ signaling can also stabilize NFκB inhibitor alpha (NFKBIA, also known as IκBα), which is a potent inhibitor of NFκB, but this mechanism has not been described for chondrocytes yet [127]. Another possible mechanism is the competition between SMAD3 and NFκB for co-activating transcription factors like CREB binding protein, as has been described in endothelial cells [128]. Additionally, many inflammatory genes counteracted by TGFβ contain both SMAD3 and NFκb binding sites, leaving room for epigenetic regulation as a mechanism.

An especially important role for TGFβ signaling lies in regulation of chondrocyte hypertrophy. TGFβ-induced pSMAD2/3 signaling blocks hypertrophy and terminal differentiation of chondrocytes [31,33,35,100,129]. In contrast, TGFβ-induced pSMAD1/5 via ALK1 is associated with chondrocyte hypertrophy [99,100]. Both effects of TGFβ on hypertrophy can be observed in vivo. Removal of TGFβ signaling in cartilage of two-week-old *Col2-CreER;Tgfbr2^flox/flox^* mice results in an OA-like phenotype within 6 months via RUNX2-mediated *Mmp13* and *Adamts5* expression [130]. Furthermore, expression of a kinase-deficient dominant negative form of *Tgfbr2* in mice induces chondrocyte hypertrophy and osteoarthritis [131]. In contrast, removal of TGFβ signaling in the hypertrophic chondrocytes of *Col10a1-Cre;Tgfbr2^flox/flox^* mice delays terminal differentiation in the epiphyseal plate [132], and removal of TGFβ signaling in cartilage of eight-week-old *AgcCreERT2^+/−^;Tgfbr2^flox/flox^* mice results in less hypertrophic chondrocytes after 12 months [133]. Because efficient removal of TGFBR2 was demonstrated in both the *Col2-CreER;Tgfbr2^flox/flox^* and *AgcCreERT2^+/−^;Tgfbr2^flox/flox^* models, their different outcomes illustrate that age of TGFβ signaling depletion (2 vs 8 weeks) and, thus, chondrocyte maturation status greatly affect the outcome and impact of TGFβ signaling in chondrocytes.

Other family members which can induce (chondro-protective) SMAD2/3 signaling in chondrocytes are the activins. However, little is known regarding their role and importance. Activin A, activin AB, and activin B signal via ACVR2A/ACVR2B and ALK4/ALK7 to induce SMAD2/3 [134]. Expression of the activin A subunit inhibin β A (*INHBA*) and activin A itself are upregulated in OA cartilage [135,136] but also expression of its inhibitor follistatin is upregulated [137,138]. A possible role, which has been demonstrated in vitro, is the inhibition of IL-1-induced ADAMTS4/5 activity [136], which would indicate an anti-inflammatory role similar to the TGFβs. However, a lack of response of adult cartilage tissue to activin A has been observed in multiple studies [109,139,140]; for example, activin A slightly induces proteoglycan synthesis (i.e., ^35^S-incorporation) in immature (1–3 months old) but not in mature (>18 months old) bovine articular cartilage explants [140]. Even less is known regarding the role in cartilage of the activin antagonists, the inhibins, and only one study reports that inhibin inhibits both proteoglycan synthesis (i.e., ^35^S-incorporation) and DNA synthesis in bovine chondrocytes [140], which would indicate that activin signaling contributes to both these processes, again similar to the TGFβs.

### 3.2. Bone Morphogenetic Proteins (BMPs)

Multiple BMPs play an important role in chondrocyte biology, including BMP2, BMP7 (OP1), BMP4, BMP6, BMP9 (GDF2), and BMP14 (GDF5) [141,142]. Chondrocytes express the BMP type I receptors ALK1, ALK2, ALK3, and ALK6 and the type II receptors BMPR2, ACVR2A, and ACVR2B. Genetic variation in either of these factors is not yet associated with OA. Like the TGFβs, BMPs are produced as a dimer, but not only as homodimers; heterodimers of BMP2/6, BMP2/7, and BMP4/7 have been described both in vitro and in vivo [143]. In contrast to TGFβ activity, BMP activity is not thought to be controlled via reassembly of BMPs with their prodomain to form a latency complex [143]. BMPs do reassemble with their prodomain to facilitate ECM binding, but this does not seem to confer latency because it does not impair receptor type I and II binding [89,143,144]. Instead, antagonistic scavenger proteins (e.g., gremlin, noggin, and sclerostin) and decoy receptors like BAMBI play an important role in modulating BMP activity, all of which are produced by chondrocytes. Additionally, BMPs can antagonize each other via competition for receptor type II binding [145].

In both healthy and OA cartilage, *Bmp2* mRNA expression can be detected [146]. BMP2 signaling induces ECM production and proliferation. In ex vivo juvenile cartilage, BMP2 induces proteoglycan content and COL2A1 production, as measured by ^3^H-proline incorporation assays [147], whereas in vivo intra-articular injection of BMP2 in murine knee joints enhances proteoglycan synthesis 2.5-fold (i.e., ^35^S-incorporation) in patellar cartilage within two days [102]. Furthermore, intra-articular overexpression of BMP2 in the murine knee joint using adenoviruses also enhances cartilage proteoglycan content [148]. In in vitro mesenchymal stromal cells and chondrocyte-like cell lines, and in in vivo epiphyseal chondrocytes, BMP2 is a well-known inducer of chondrocyte maturation and hypertrophy [149]. However, in mice with cartilage-specific, tamoxifen-inducible overexpression of BMP2 (*Col2 rTA; TRE-Bmp2*), increased hypertrophy is not observed in articular cartilage after six weeks of prolonged exposure to tamoxifen [150]. Furthermore, in the DMM model of experimental OA, the presence of excessive BMP2 signaling does not negatively, nor positively, affect cartilage damage but does induce very extensive osteophyte formation [150]. These observations might indicate that, in vivo, articular cartilage is not very sensitive to BMP2 signaling compared to osteophyte-forming tissues like periosteum. Possibly, signaling of hyaluronan, an important structural component of cartilage, via CD44 plays a role in this, as it has been demonstrated that this inhibits BMP2-induced pSMAD1/5 signaling in the murine bone marrow-derived ST2 stromal cell line [151]. Alternatively, GDF5 has recently been shown to be a context-dependent inhibitor of BMP2 signaling, via a yet unknown mechanism, possibly involving receptor competition [152], and GDF5 is present in cartilage [153,154,155].

*BMP7* mRNA and protein expression can be detected in both immature and mature articular cartilage. Immunohistochemically, mature BMP7 can predominantly be detected in chondrocytes of the superficial zone, whereas its pro-form is mainly detected in deep zone chondrocytes [156,157]. To signal, BMP7 can use ALK2, ALK3, and ALK6, and the type II receptors BMPR2, ACVR2A, and ACVR2B, to induce SMAD1/5 phosphorylation and SMAD-independent signaling [143]. In contrast to BMP2, a positive interaction of CD44 with BMP7 signaling has been described. Without CD44, chondrocytes are not able to induce pSMAD1/5 in response to BMP7 [158,159,160]. In chondrocytes, BMP7 signaling induces ECM production. Both proteoglycan and *COL2A1* production are induced in vitro in juvenile and adult human primary chondrocytes [139] and ex vivo in juvenile bovine explants [147]. Furthermore, inhibition of endogenous BMP7 production with antisense oligonucleotides in cartilage explants lowers aggrecan production [161]. In vitro, BMP7 also induces the expression of cartilage oligomeric matrix protein [109] and hyaluronan synthase 2, resulting in more hyaluronan production by chondrocytes, possibly facilitating deposition of aggrecan in the ECM [162]. In addition, BMP7 can counteract an IL-1- or LPS-induced inhibition of proteoglycan synthesis in chondrocytes [163,164,165], and partly inhibit IL-1-induced *MMP13* expression, especially in the presence of IGF1 [166]. The positive effects of BMP7 in vitro are reflected in vivo: conditional deletion of *Bmp7* from limb mesenchyme in *Prx-Cre;Bmp7^flox/flox^* mice results in proteoglycan loss (at 8 weeks old) and enhanced *Mmp13* expression (at 24 weeks old) [167]. Of note, joint and cartilage formation were not detectably affected in these mice, but *Bmp7* deletion did result in synovial inflammation, making it difficult to exclude indirect effects. Administration of BMP7 is beneficial for cartilage in OA [168,169]. Osteoarthritic cartilage is still responsive to BMP7 [155], and weekly injections of BMP7 diminish cartilage damage in the rabbit ACLT model of OA [170]. Furthermore, BMP7 increases the quantity and quality of cartilage repair tissue in rabbits with a full thickness cartilage defect [171]. Remarkably, although BMP7 can induce ectopic bone formation [172], excessive osteophyte formation was not observed in these studies, nor was excessive osteophyte formation observed in humans after a single injection of BMP7 in a Phase 1 study done in symptomatic knee OA patients [173]. Although BMP7 induces SMAD1/5 phosphorylation, its signaling is not associated with chondrocyte hypertrophy. Bovine articular chondrocytes do not undergo hypertrophy when cultured with BMP7 [174], and BMP7 blocks hypertrophy in the ATDC5 cell line [175]. In long bone formation ex vivo, BMP7 does stimulate the formation of pre-hypertrophic chondrocytes in the epiphyseal plate but blocks the transition towards hypertrophy [176]. This anti-hypertrophic effect of BMP7 has contributed to its ability to induce the transcription factor BAPX-1/NKX-3.2, a potent modulator of chondrocyte hypertrophy [177]. However, BMP7 does induce ALPL activity and osteogenesis in primary fetal chondrocytes [178], and chick sternum chondrocytes do undergo hypertrophy when cultured with BMP7 [179], indicating that BMP7 can induce chondrocyte hypertrophy, but, like TGFβ, it is in a context-dependent manner.

Two other BMPs well known for their positive effect on chondrogenesis and cartilage formation are BMP4 and BMP6, which both signal via ALK2 and ALK3 to induce pSMAD1/5 and SMAD-independent signaling. However, very little is known regarding their role in mature articular cartilage. In bovine chondrocytes and cartilage explants, BMP4 enhances proteoglycan synthesis, COL2A1 production, and proliferation [140], and BMP4 has similar effects in human chondrocytes cultured in alginate [164]. BMP6 mRNA and protein are expressed in both healthy and OA articular cartilage [141]. In human chondrocytes, BMP6 induces proteoglycan synthesis [164,180], but with advancing age, chondrocyte response to BMP6 decreases [180]. However, in the epiphyseal plate, BMP6 is especially expressed in the hypertrophic zone [181] where it has a stimulatory role on chondrocyte maturation. Furthermore, a pro-hypertrophic effect of BMP6 has also been observed in ATDC5 cells [182]. These studies indicate that BMP6 signaling could be detrimental for articular cartilage

A BMP that has more recently come to attention is BMP9, also known as GDF2. This BMP signals via type II receptors ACVR2A/ACVR2B and BMPR2 and type I receptors ALK1, and/or ALK2 to induce SMAD1/5 and SMAD-independent signaling. Production of BMP9 could not be detected in chondrocytes [141], but BMP9 circulates in high levels in blood [183]. In juvenile bovine cartilage, BMP9 enhances the production of proteoglycans and does so more potently than BMP2 [147]. However, BMP9 can also induce ECM mineralization when chondrocytes are seeded in biodegradable polyglycolic acid (PGA) scaffolds [184]. Additionally, also in mesenchymal progenitor cells, BMP9 is a potent inducer of chondrogenesis accompanied by hypertrophy [185]. Therefore, BMP9 possibly has a detrimental effect on chondrocyte homeostasis. Indeed, increased hypertrophy was observed in primary bovine chondrocytes stimulated with this BMP, but remarkably, this was efficiently blocked via a synergy with TGFβ1 on SMAD2/3 phosphorylation [186].

BMP14, better known as GDF5, is a very important initiator of joint formation and patterning. The importance of GDF5 in cartilage biology is illustrated by the observations that mutations in *GDF5* can lead to chondrodysplasias, and that a genetic SNP in the core promotor of *GDF5,* resulting in reduced GDF5 protein levels, is associated with OA development [187]. However, regarding the role of GDF5 in mature cartilage, relatively little is known. In chondrocytes, GDF5 predominantly signals via ALK6 [178] but can also signal via ALK2 and ALK3 [152] to induce SMAD1/5 phosphorylation and SMAD-independent signaling [188]. *GDF5* is expressed in (im)mature cartilage, but its expression is not affected by OA [141,154,155]. In cartilage, GDF5 induces proteoglycan synthesis [154,178] and *COMP* expression [109] but neither *COL2A1* synthesis nor chondrocyte proliferation [154]. Notably, OA chondrocytes do not respond consistently to GDF5 [188]; that is, large variations can be observed between OA patients on gene expression of targets like *ACAN* and *SOX9* after addition of GDF5. Possibly, a yet undiscovered co-receptor plays a role in this. Such a co-receptor has been postulated to explain how GDF5 can function as a downstream BMP2 antagonist in ATDC5 cells even though it uses the same type I and type II receptors [152]. Identification of such a co-receptor would greatly help in understanding GDF5 signaling in chondrocytes and how it differs from BMP signaling using the same receptors.

Figure 4 summarizes the effects of the different TGFβ family members on cartilage biology and OA development.

## 4. Changes in TGFβ Family Signaling as Cause for Disease

### 4.1. Age-Related Changes in TGFβ Signaling

In view of the aforementioned importance of TGFβ family members in chondrocyte and cartilage homeostasis, it is not surprising that alterations in their signaling affect cartilage biology and can lead to pathology. Alterations can occur because of various reasons, for example due to aging. Aging affects TGFβ family signaling on multiple levels in the signaling cascade. For example, ligand and receptor expression can be affected, just as intracellular signaling and regulation of gene transcription [90,121,189,190,191].

A prime example of how aging affects ligand expression is BMP7. In human cartilage, BMP7 gene and protein levels decline with advancing age. This decrease has been attributed to increased methylation of its promoter, resulting in silencing of its expression [192,193]. In contrast, mRNA expressions of *BMP2*, *BMP6,* and *BMP9* do not seem to be affected by aging [141,146,156,190]. In mice, protein expression levels of TGFβ2 and TGFβ3 (but not TGFβ1) decreased with age [90], whereas in cows, a drop in *Tgfb1* mRNA expression was observed [142].

Also, receptor expression is impacted by aging. Of type I receptors, a profound drop in ALK5 expression has been observed in humans, mice, guinea pigs, and cows on gene expression and protein level [100,142,194], indicating that this occurs consistently across species. In addition, on gene expression levels, loss of *Alk2*, *Alk3,* and *Alk4* has also been observed in bovine cartilage [90,142]. Notably, loss of receptor expression can not only lead to loss of the growth factor response but also alters the signaling of ligands able to use multiple receptors. For example, this has been demonstrated for TGFβ signaling because aging affects *Alk5* expression more than *Alk1* expression, and TGFβ induces relatively more SMAD1/5 phosphorylation in old cartilage, with all due effects [142,194]. Next to these differences in type I receptor expression, an age-related decrease in *Bmpr2* has also been observed [142]. This could possibly explain the age-related loss in the BMP7 response of chondrocytes that has been observed [190]. However, how age-related changes affect BMP signaling (besides BMP7) in chondrocyte homeostasis has not yet been sufficiently investigated.

Changes in receptor expression can affect intracellular signaling. As mentioned, an age-related change in receptor type I expression towards relatively more ALK1 shifts the balance from protective signaling (SMAD2/3) into deleterious signaling (SMAD1/5/9). The loss of protective SMAD2/3 signaling could explain the relation between aging and OA development [12,195]. Indeed, the changing ALK1/ALK5 ratio in aging correlates with *Mmp13* expression and OA development [89]. Computational modeling supports a direct effect, especially TGFβ signaling via ALK1, which can explain *Mmp13* expression and cartilage damage in aged cartilage [196]. Possibly, these effects run via decreased modulation of RUNX2 activity. SMADs interact with RUNX2 to modulate chondrocyte differentiation, where interaction of SMAD3 with RUNX2 inhibits RUNX2 functioning and SMAD1/5 stimulates it [36,53,197,198].

### 4.2. Joint Loading-Related Changes in TGFβ Signaling

Next to aging, another factor able to modulate TGFβ family signaling is (mechanical) loading. For example, mechanical joint loading has been shown to play a crucial role in activation of SMAD2/3 signaling [97,199,200]. In bovine articular cartilage explants, mechanical compression quickly induced SMAD2/3 signaling, most likely due to mechanical activation of latent TGFβ bound to the ECM of cartilage, as cartilage contains high levels of this. This loading-induced pSMAD2/3 signaling induced a positive feedback loop, by stimulating expression of *Tgfb1* and *Alk5*. In contrast, unloading rapidly induced loss of SMAD2/3 signaling from articular cartilage (<2 h), possibly due to sequestering of available TGFβ [201]. As a consequence of this, prolonged (2 weeks) unloading of cartilage resulted in dramatically increased *Col10a1* expression, a marker of chondrocyte hypertrophy. However, repeated loading of cartilage every third day efficiently blocked this increase in *Col10a1* expression. These observations show the importance of cartilage loading for maintenance of SMAD2/3 signaling and cartilage homeostasis. In view of the effects of SMAD3 signaling in cartilage, this mechanism possibly also explains the observations that reduced joint loading contributes to cartilage degeneration and production of the proteolytic enzymes MMP8 and MMP13 [202,203]. Importantly, old cartilage, compared to young cartilage, has a strongly reduced capacity for this loading-mediated SMAD2/3 signaling, which possibly makes it more vulnerable to OA development [201,204]. Interestingly, a sedentary lifestyle was recently hypothesized to be the cause for the doubled OA prevalence in current populations compared to those from the 1800s to early 1900s after correction for age and BMI [205]; thus, possibly these observations are linked. Loading also induces BMP levels; BMP2, BMP4, BMP6, and BMPR2 are increased by moderate exercise and suppresses post-traumatic OA [206]. However, excessive loading has been shown to lead to collagen damage, proteoglycan loss, and OA progression, and these effects have been attributed to induction of gremlin-1, a BMP antagonist [207,208]. Together, these studies show the impact of loading on TGFβ family signaling.

### 4.3. Inflammation-Related Changes in TGFβ Signaling

Previously, cartilage breakdown during OA was considered to be a simple wear and tear process. However, nowadays, OA is regarded as a whole joint disease, with inflammation strongly implicated in its pathogenesis [209,210]. In over 50% of patients, synovitis (i.e., inflammation of the synovium or joint capsule) can be observed [211,212,213,214,215,216,217]. Important inflammatory mediators produced by this synovitis are IL-1, TNF-α, IL-8, and IL-6 [210,218,219,220,221]. These pro-inflammatory cytokines have a direct effect on cartilage homeostasis by upregulating MMPs and other cartilage-degrading enzymes [222,223,224,225] but, importantly, also have been described to modulate TGFβ family signaling [226,227].

For example, IL-1 can downregulate TGFBR2 on mRNA and protein expression in human articular chondrocytes [226]. Also, pro-inflammatory mediators produced by inflamed synovium, which was cultured in vitro (OA synovium conditioned medium), reduced *Tgfbr2* expression in mechanically compressed bovine cartilage explants [228]. IL-1 has also been described to inhibit BMP7 signaling via downregulation of the ALK2 and ALK3 receptors and inhibition of *SMAD1* expression [229]. But in contrast, IL-1 has also been described to induce *BMP2* and *BMP7* mRNA expression in chondrocytes, and a high dose of IL-1 leads to the activation of pro-BMP7 into the active form [230].

Next to modulation of receptor expression, inflammatory mediators can also affect SMAD-dependent signaling via various mechanisms. For instance, IL-1 can increase the expression of i-*SMAD7* via NF-κB activation in chondrocytes, which inhibits SMAD2/3 signaling [231]. Furthermore, TNF-α and IL-1 have been described to be able to decrease DNA binding activity of SMAD3/4 complexes in human OA chondrocytes via unknown mechanisms but independent of i-SMAD7 [227]. In addition, inflammation-induced transcription factors can compete with SMADs for co-factors. For example, SMAD3 and NFκB (a transcription factor downstream of TAK1) can compete for their common co-transcription factor CREB binding protein (CBP) in endothelial cells [127].

Recent insights provided another way of how pro-inflammatory pathways can interact with SMAD-dependent signaling, namely by post-translational modification of the linker region of SMAD proteins [232,233]. SMAD proteins consist of a N-terminal MH1 (missing in i-SMAD6 and i-SMAD7), important for DNA binding activity and nuclear importation, and a C-terminal MH2 domain, responsible for SMAD-receptor and SMAD–SMAD interactions and gene transcription activation [234,235,236,237]. These two conserved MH1 and MH2 domains are connected via a regulatory linker domain, which can be phosphorylated on specific serine and threonine residues. This phosphorylation of the linker region regulates nuclear entry, SMAD–protein interactions, and SMAD turnover [232,238,239]. For instance, mutation of the *SMAD3* linker resulted in strongly enhanced TGFβ-induced responses in breast cancer cells and increased tumorigenesis during liver cancer [240,241]. Also, mutations in the linker of *SMAD4*, commonly found in colorectal and pancreatic cancers, result in rapid proteasomal degradation and loss of function [242,243]. These findings show that modification of the SMAD linkers greatly affect SMAD function (Figure 3).

Phosphorylation of the serine and threonine residues in the linker domain is dependent on kinases, including MAPKs, ERK, JNK, p38, AKT, CDKs, ROCK, and GSK-3 [233,238,239,244,245,246]. In addition, dephosphorylation of the SMAD linker, dependent on phosphatases [237,247,248], is just as important for regulating SMAD function. Notably, many of these kinases and phosphatases are induced in chondrocytes by inflammatory mediators [222,249,250,251,252] present in the OA joint. This makes it likely that inflammation-induced kinases can also affect SMAD function in cartilage, thereby affecting TGFβ and BMP signaling and, thus, a possible cause for OA. Still, the relative importance of these SMAD linker modifications in cartilage biology and OA pathogenesis is not well understood yet and warrants further research.

A (simplified) overview of SMAD modifications in the linker-domain that can alter TGFβ signaling is depicted in Figure 5.

## 5. Future Perspectives

In conclusion, members of the TGFβ family are crucial for the homeostasis of chondrocytes and, thus, of cartilage. Signaling in cartilage goes via both SMAD-dependent and SMAD-independent signaling, although both pathways are more integrated than their names suggest. Cellular context plays a large role in TGFβ family signaling, greatly affecting its outcome, which is illustrated by, for example, the differential effects of TGFβ and BMP7 in immature versus mature chondrocytes. How aging or inflammation provide cellular context, which regulates TGFβ family signaling in articular cartilage, is still largely unknown. Gaining greater insights into this would help to better understand OA development and progression, and it would give leads for how to separate the beneficial effects of TGFβ family members from their detrimental effects. This topic, thus, definitely warrants further research. Of special interest might be the interaction of SMADs with other transcription factors. Such interactions control (efficient) targeting and transcription of (potential) target genes. Altered interactions, for example due to altered SMAD linker modifications, could explain context-dependent gain and loss of function. How, for instance, the SMAD–RUNX2 interaction is affected by aging or inflammation could be of great importance for cartilage degeneration. Closely related to this, it would be important to further study other SMAD binding partners involved in regulating chondrocyte hypertrophy such as myocyte enhancer-binding factor 2 (MEF2). This transcription factor has been shown to induce RUNX2 expression and hypertrophy in cartilage [253], and it has been described to interact and be inhibited by SMAD3 in muscle cells [254,255]. Identifying such interactions in articular chondrocytes can help us learn how cartilage degradation is regulated and would give leads on how to counteract this. Another topic of interest is preventing the negative impact of processes such as inflammation on TGFβ family signaling in cartilage. For this, a better understanding of the cellular processes driving these negative effects is needed, but this hardly has been investigated until now. Especially, the relevance of inflammation-driven SMAD linker post-translational modifications as causes for disturbed TGFβ and BMP signaling warrants attention, because this possibly will provide a new avenue of targeted therapy directed at the inhibition of cartilage degradation in OA by targeting the intracellular kinases and phosphatases responsible for these modifications.

## Figures and Tables

**Figure 1 cells-08-00969-f001:**
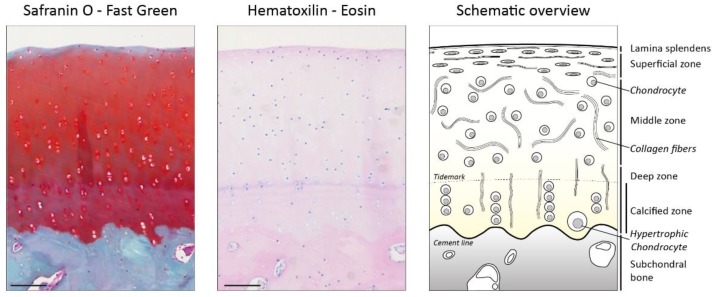
Appearance and structure of adult (bovine) articular cartilage. Note that the orientation of collagen fibrils differs per zone, from parallel in the superficial zone to perpendicular in the deep zone. Scale bar = 100 µm.

**Figure 2 cells-08-00969-f002:**
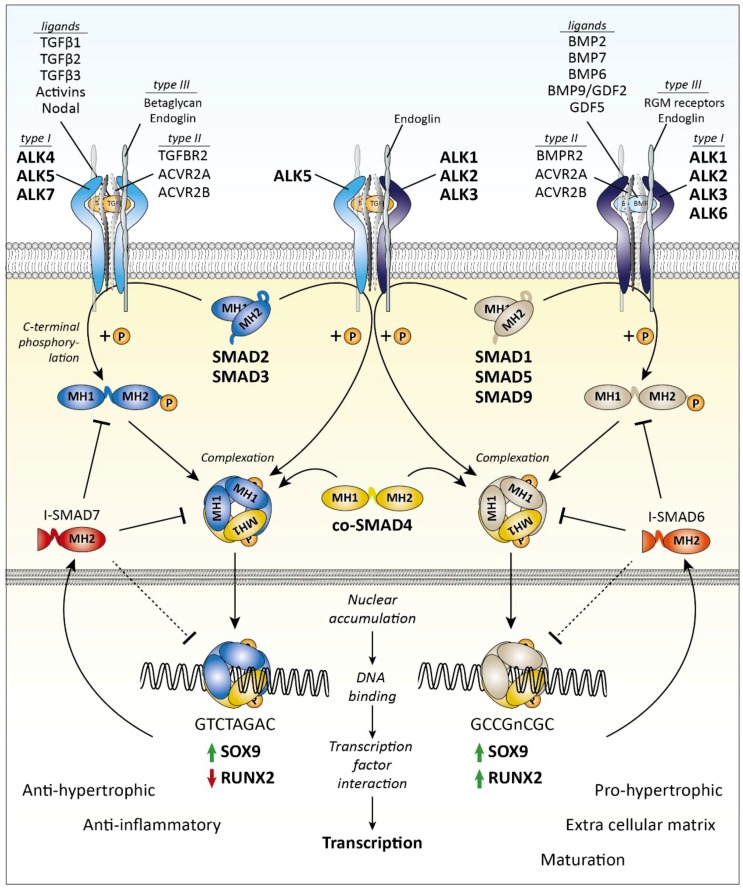
Transforming growth factor β (TGFβ) family SMAD-dependent signaling. Ligand binding induces the formation of a receptor complex containing a type I and type II receptors. Type III receptors stabilize and facilitate the binding of TGFβ to the receptor complex, which recruits a receptor SMAD (R-SMAD). The R-SMAD is subsequently phosphorylated on its C-terminal domain, and a complex is formed with the common SMAD4 (co-SMAD4). This complex translocates to the nucleus where it can bind transcription factors, like SOX9 and RUNX2, and activates transcription. The activation of SMAD2/3 signaling results in anti-hypertrophic and anti-inflammatory function, whereas activation of SMAD1/5/9 signaling is associated with pro-hypertrophic regulation of the extracellular matrix and maturation of the cartilage. Other important target genes are the inhibitory SMADs, *SMAD6* and *SMAD7,* whose expression provides the cell with a negative feedback mechanism. ACVR2 = Activin type-2 receptor, ALK = ALK tyrosine kinase receptor, BMP = Bone morphogenetic protein, BMPR = Bone morphogenetic protein receptor, GDF2 = Growth differentiation factor 2, MH(1/2) = MAD homology (1/2) domain, RGM = Repulsive guidance molecule, RUNX2 = Runt-related transcription factor 2, SOX9 = Transcription factor SOX9, TGFBR = TGFβ receptor

**Figure 3 cells-08-00969-f003:**
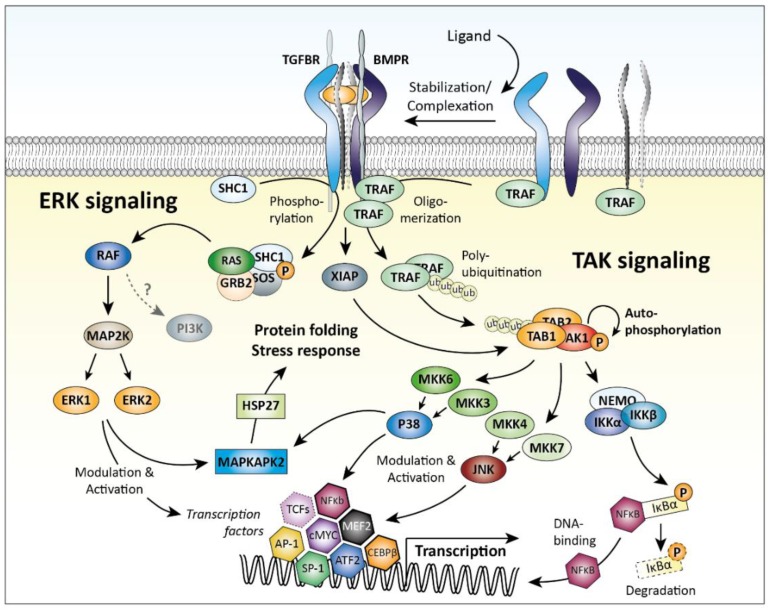
Transforming growth factor β (TGFβ) family SMAD-independent signaling. Ligand binding induces the formation of a receptor complex. This complex can initiate SMAD-independent signaling via activation of adaptor proteins, e.g., SHC1 and TRAF6. This activation results in initiation of the ERK and TAK signaling cascades, culminating in the activation of MAPKAPK2, p38, JNK, and NFκb. These pathways induce transcription and translation and regulate diverse cellular processes. AKT = RAC-alpha serine/threonine-protein kinase, AP-1 = Activator protein 1, ATF2 = Cyclic AMP-dependent transcription factor ATF-2, CEBPβ = CCAAT/enhancer-binding protein β, ERK = Mitogen-activated protein kinase 3, GRB2 = Growth factor receptor-bound protein 2, HSP27 = Heat shock protein beta-1, IkBa = NF-kappa-B inhibitor alpha, IKK = Inhibitor of nuclear factor kappa-B kinase, JNK = c-Jun N-terminal kinase, MAP2K = Dual specificity mitogen-activated protein kinase kinase 1, MAPKAPK = MAP kinase-activated protein kinase, MEF2 = Myocyte-specific enhancer factor 2A, MKK = Dual specificity mitogen-activated protein kinase kinase, cMYC = Myc proto-oncogene protein, NEMO = NF-kappa-B essential modulator, NFκb = nuclear factor kappa-light-chain-enhancer of activated B cells, p38 = p38 mitogen-activated protein kinases, PI3K = Phosphatidylinositol 3-kinase, RAF = RAF proto-oncogene serine/threonine-protein kinase, RAS = GTPase HRas, SHC1 = SHC-transforming protein 1, SOS = Son of sevenless homolog 1, SP-1 = Sp1 transcription factor, TAB = TGF-beta-activated kinase 1 and MAP3K7-binding protein 1, TAK1 = TGFβ-activated kinase 1, TCFs = ternary complex factors, TRAF6 = TNF receptor-associated factor 6, and XIAP = E3 ubiquitin-protein ligase XIAP.

**Figure 4 cells-08-00969-f004:**
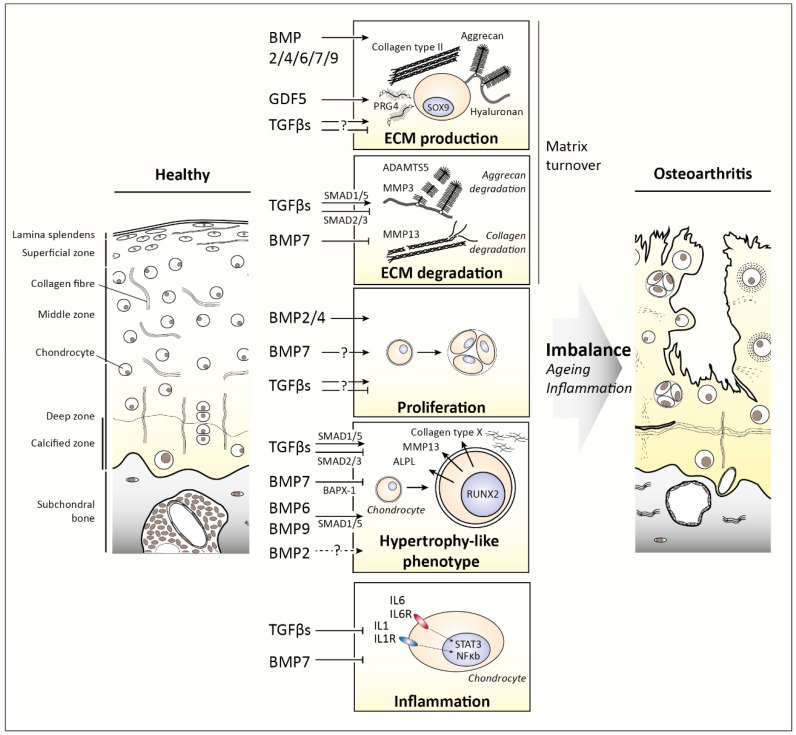
Effects of the different Transforming Growth Factor β (TGFβ) family members on articular cartilage biology and osteoarthritis (OA) development. On the left healthy articular cartilage is depicted, on the right osteoarthritic cartilage. The middle depicts how TGFβ family members regulate matrix turnover, chondrocyte proliferation, hypertrophy-like differentiation, and inflammation. An imbalance in these effects of TGFβ family members because of age- and/or inflammation-related changes contributes to OA development. ADAMTS5 = A disintegrin and metalloproteinase with thrombospondin motifs 5, ALPL = Alkaline phosphatase tissue-nonspecific isozyme, BAPX-1 = Homeobox protein Nkx-3.2, BMP = Bone morphogenetic protein, GDF = Growth differentiation factor, IL = interleukin, MMP = matrix metalloproteinase, NFκb = nuclear factor kappa-light-chain-enhancer of activated B cells, PRG4 = proteoglycan 4, RUNX2 = Runt-related transcription factor 2, SOX9 = Transcription factor SOX-9, and STAT3 = Signal transducer and activator of transcription 3.

**Figure 5 cells-08-00969-f005:**
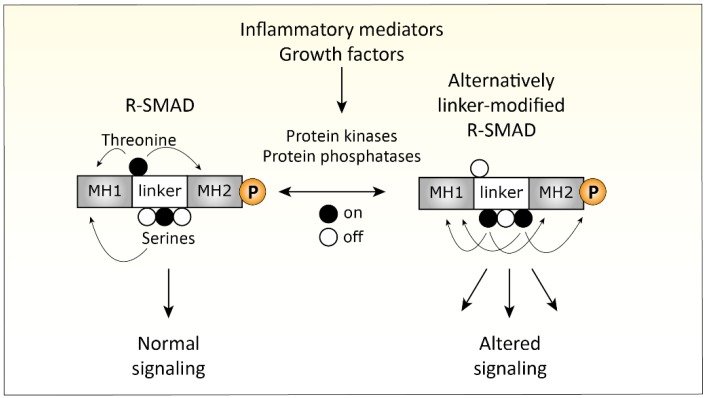
SMAD linker modifications alter Transforming growth factor β (TGFβ) signaling. The SMAD proteins consist of a MH1 and MH2 domain, connected with a non-conserved linker domain. Threonine and serine residues in the linker domain can be modulated by protein kinases and phosphatases like mitogen-activated protein kinases (MAPKs), c-Jun N-terminal kinases (JNK), and Glycogen synthase kinase 3 (GSK-3). These kinases can be activated by growth factors and inflammatory mediators. This process greatly affects SMAD function and leads to altered TGFβ signaling.

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
