# Peer review of "TGFβ/BMP Signaling Pathway in Cartilage Homeostasis"

_cells, 2019, doi:10.3390/cells8090969_

Round 1
Reviewer 1 Report
The authors comprehensively review the current knowledge of the TGFb and BMP signalling pathways in the homeostasis of chondrocytes in humans and other species. It is supported by the published in vivo and in vitro experimental evidence and the implications in health and disease states appropriate for current generation of ageing population are discussed. The review is well written and easy to follow.
I have noticed few punctuation mistakes and perhaps misleading use of capitalised letters and smaller case for genes and proteins, particularly in the later parts of the manuscript. Please check that.
Author Response
Dear reviewer 1,
Thank you for taking the time and effort to read and review our paper, and thank you for your compliments regarding its quality.
Comment: I have noticed few punctuation mistakes and perhaps misleading use of capitalized letters and smaller case for genes and proteins, particularly in the later parts of the manuscript. Please check that.
Response: Thank you for noticing these errors. We are sorry for this, and have checked the manuscript on nomenclature mistakes and made numerous adjustments to make sure that we follow the HUGO (human) and MGI (mouse) nomenclature convention. This means that human genes are written in capitals and in italic, e.g. SMAD3, and human proteins in capitals, e.g. SMAD3 (HGNC Guidelines | HUGO Gene Nomenclature Committee), whereas murine genes are written with the first letter as a capital and the whole name in italic, e.g. Smad3, but the protein is written in capitals: SMAD3 (MGI-Guidelines for Nomenclature of Genes, Genetic Markers, Alleles, & Mutations in Mouse & Rat). Hopefully, there are now no mistakes left.
Reviewer 2 Report
This is a very well written, comprehensive review on TGFb/BMP signalling in cartilage biology. The molecular pathways and interactions between different molecules are described with great detail and interesting thoughts.
This review would benefit from broadening the topic from cartilage only to joint tissues since the main relevance to disease is osteoarthritis, which is a multi-tissue disease rather than disease of cartilage only.
Future perspectives section is short, perhaps this could be expanded as well.
Author Response
Dear reviewer 2,
Thank you for taking the time and effort to read and review our paper. We have tried to address your comments as follows:
Comment 1: This review would benefit from broadening the topic from cartilage only to joint tissues since the main relevance to disease is osteoarthritis, which is a multi-tissue disease rather than disease of cartilage only.
Response: Thank you for this comment. We fully agree with you that osteoarthritis is a multi-tissue disease rather than a disease of the cartilage only. We also think that the TGFβ family plays a role in OA pathophysiology in tissues such as synovium, meniscus and bone. However, we have chosen to limit the scope of our review to cartilage alone, because we were invited to specifically write about cartilage, and cartilage degradation is an irreversible hallmark of OA pathology in which the TGFβ family clearly plays a role. It is of course very interesting to discuss the involvement of the TGFβ family in other tissues in osteoarthritis pathophysiology as well, but this would be an excellent topic for a different review. In the current review we did include more detailed reasons why we focus on articular cartilage, and added some more background information on cartilage structure, development and homeostasis in general in lines 28 to 100, including an extra figure 1.
Comment 2: Future perspectives section is short, perhaps this could be expanded as well.
Response: We indeed do agree with this comment. Many unanswered questions remain and of course much more can be done in future research. Therefore, we expanded this paragraph with lines 1082 to 1104. Here we describe how important cellular context is for regulating TGFβ family signaling, since TGFβ family members can both have beneficial and detrimental effects on articular cartilage. How ageing and inflammation provide cellular context is still largely unknown and is of great importance for future research. For instance, gaining insights in how ageing or inflammation affect the interaction of SMAD-dependent signaling with transcription factors like RUNX2, would learn us how cartilage degradation is regulated. Another topic of interest is preventing the negative processes such as inflammation on TGFβ family signaling in cartilage, for instance by targeting intracellular kinases and phosphatases, which are responsible for SMAD linker modifications.
Reviewer 3 Report
This manuscript reviews literature on TGF/BMP signaling in cartilage homeostasis.
The authors appear to focus on articular cartilage homeostasis. The review does not provide an overview of the biology of articular cartilage development and homeostasis. Articular cartilage and growth plate cartilage are very different in the chondrocyte lineage, chondrocyte behavior, matrix composition, etc. For readers to better understand the topics of the review, the authors should provide a paragraph first discussing articular cartilage development and homeostasis in general, including key molecules related to the topic.
2. Different types of experimental systems (in vitro, in vivo, growth plates, and articular chondrocytes) are often mixed to support statements. The authors should primarily focus on the findings in articular cartilage/chondrocytes.
3. Related to 2. It would be very helpful for the readers to comrehensively understand this topic if a summary table (and/or figure) is included.
Minor issues:
4. ln 116. Runx2's stimulatory role of hypertrophy was first demonstrated in vivo by the Karsenty's group (Takeda S et al. Genes Dev. 2001)
5. ln 127. Cartilage-specific Smad 2 KO. This statement is incorrect. Please check the original paper.
6. Gene nomenclature should be in accord with MGI (mouse) and HUGO (human). e.g. Smad3-null mice, but not SMAD3-null mice
7. ln 150. Stimulatory role of Runx2 in hypertorphy. It is considered now that the major transcriptional driver of hypertrophy is Mef2. In relation with chondrocyte hypertrophy, SIK/HDAC/MEF2 pathway should be also discussed (e.g. PMID 17336904; 27009967)
8. ln 303. Tgfbr2 conditional KO is not performed in the literature 118.
Author Response
Dear reviewer 3,
Thank you for taking the time and effort to read and review our paper. We have tried to address your comments as follows:
Comment 1: The authors appear to focus on articular cartilage homeostasis. The review does not provide an overview of the biology of articular cartilage development and homeostasis. Articular cartilage and growth plate cartilage are very different in the chondrocyte lineage, chondrocyte behavior, matrix composition, etc. For readers to better understand the topics of the review, the authors should provide a paragraph first discussing articular cartilage development and homeostasis in general, including key molecules related to the topic.
Response: Thank you for this suggestion. Because not all readers will originate from a cartilage research field, we definitely agree that our review would benefit from a better introduction in articular cartilage structure and development and explanation why we focus on this key topic in this review. We added line 28 to 100 to better introduce articular cartilage and biology and an extra figure (figure 1).
Comment 2: Different types of experimental systems (in vitro, in vivo, growth plates, and articular chondrocytes) are often mixed to support statements. The authors should primarily focus on the findings in articular cartilage/chondrocytes.
Response: Thank you for this comment. Where possible, we used findings in articular cartilage and chondrocytes. We agree with you that for growth plate cartilage is different from articular cartilage, and that processes in these tissues are not necessarily the same (but also not necessarily different). We have therefore tried to limit the use of these findings as much as possible. However, data of articular cartilage is not always available, especially of various knockout animals. In these cases we have tried to combine in vivo observations in the growth plate with in vitro data of articular chondrocytes to strengthen conclusions.
Comment 3: Related to 2. It would be very helpful for the readers to comprehensively understand this topic if a summary table (and/or figure) is included.
Response: Thank you for this suggestion. We agree that it is a good idea to add a summary figure, this will definitely improve the paper. Therefore we have made figure 4. This figure depicts the roles of various TGFβ family members in key processes in cartilage homeostasis.
Minor issue 4: ln 116. Runx2's stimulatory role of hypertrophy was first demonstrated in vivo by the Karsenty's group (Takeda S et al. Genes Dev. 2001)
Response: Thank you for this remark, this reference (nr. 38) had been added to the manuscript.
Minor issue 5: ln 127. Cartilage-specific Smad 2 KO. This statement is incorrect. Please check the original paper.
Response: We have checked the original paper, but we are not sure why you do not agree with this statement. In line 127 we refer to [35] which is Wang W, et al, (PMID: 27741240). In this paper three mouse models were analyzed: “Therefore, we analyzed mice in which Smad2 is deleted in cartilage (Smad2CKO), global Smad3-/- mutants, and crosses of these strains” (quote from the abstract). In all figures data is show of these 3 models; and we based sentences 127-130 on the reported observations of figure 2 and supplemental figure 4 of this paper; showing that the hypertrophic zone is elongated in the Smad2CKO mice . (To make the SMAD2 cartilage specific knockout animals in this study; Smad2fx/fx mice were intercrossed with the Col2a1-Cre deleter strain to generate Smad2fx/fx;Col2a1Cre (Smad2CKO) mice. The use of Col2a1-Cre mice makes the deletion largely cartilage specific). We did rephrase the remark of the shortened proliferative zone, as it is larger; the resting zone is shorter. What we meant was that this Smad2CKO model shows that SMAD2 inhibits growth plate chondrocyte proliferation in mice, because the proliferative zone is longer in knockout animals. In addition; we have added Smad2fx/fx;Col2a1Cre mice to the line to clarify the model
Minor issue 6: Gene nomenclature should be in accord with MGI (mouse) and HUGO (human). e.g. Smad3-null mice, but not SMAD3-null mice
Response and action: Thank you for this comment and we are sorry for these mistakes. We checked the manuscript and made several adjustments to make sure we now indeed follow the MGI and HUGO nomenclature conventions. This means that human genes are written in capitals and in italic, e.g. SMAD3, and human proteins in capitals, e.g. SMAD3 (HGNC Guidelines | HUGO Gene Nomenclature Committee), whereas murine genes are written with the first letter as a capital and the whole name in italic, e.g. Smad3, but the protein is written in capitals: SMAD3 (MGI-Guidelines for Nomenclature of Genes, Genetic Markers, Alleles, & Mutations in Mouse & Rat). Hopefully, there are now no mistakes left.
Minor issue 7: ln 150. Stimulatory role of Runx2 in hypertrophy. It is considered now that the major transcriptional driver of hypertrophy is Mef2. In relation with chondrocyte hypertrophy, SIK/HDAC/MEF2 pathway should be also discussed (e.g. PMID 17336904; 27009967)
Response: You raise an interesting point. In this paragraph we discuss SMAD dependent signaling and the role of the SMADs in hypertrophic regulation. We agree that not only RUNX2, but also other transcription factors like MEF2 are important in chondrocyte hypertrophy in articular cartilage as has been described by Arnold MA, et al, (PMID 17336904). Similarly to for RUNX2, an interaction of MEF2 with SMAD dependent signaling of TGFβ family members (i.e. SMAD2 and SMAD4) has been described by Zoë A. Quin, et al, (PMID 11160896). They show that in myoblasts, SMAD2/4 interacts with MEF2, in a p38 MAPK-dependent manner. Another paper, by Dong Liu, et al, (PMID 15044954) shows that TGFβ-activated SMAD3 represses the function of MEF2 during myogenesis, thereby silencing muscle gene expression of myogenin and displacing the GRIP-1 coactivator from MEF2 target promotors. The significance of this functional interaction between TGFβ family members and MEF2 may extend beyond skeletal muscle differentiation and play a role in articular chondrocytes as well, but the interaction of these pathways has not yet been described in articular cartilage. In addition, only little is known of MEF2 in cartilage at all, as a pubmed search using MESH terms for: MEF2 Transcription Factors AND cartilage returns only 13 hits, and MEF2 Transcription Factors AND chondrocytes only 12 hits, with none of these papers reporting on the interaction of SMADs with MEF2c. In bone (MEF2 Transcription Factors AND bone AND Smad proteins) only one paper mentions a possible interaction between SMADs and MEF2; Kawane T., et al, (PMID 24692107) describe that SMAD1 enhances MEF2 function on the RUNX2 promoter.
It would indeed be very relevant and interesting if an interaction between MEF2 and SMADs plays a role in articular cartilage too, therefore we added a sentence about this topic in the future perspectives. However, because of the lack of data regarding an interaction between MEF2 and SMADs in cartilage, in contrast to the SMAD-RUNX2 interaction which has been reported on several times , we decided to keep our paragraph only to discussion of the RUNX2-SMAD interaction and how this regulates hypertrophic differentiation .
Minor issue 8: ln 303. Tgfbr2 conditional KO is not performed in the literature 118.
Response: Thank you for noticing this mistake. Reference 118 belongs to line 306 and not to line 303. This has been corrected.
Round 2
Reviewer 3 Report
The manuscript has been significantly improved.
(Regarding the Smad2 mouse knockout paper, as the authors corrected, the paper does not say shortening of tbe proliferative zone in Smad2, but says shortening of the resting zone, and elongation of the columnar and hypertrophic zones)